# Changes in Land Plot Morphology Resulting from the Construction of a Bypass: The Example of a Polish City

Cezary Kowalczyk, Jacek Kil and Krystyna Kurowska *

Faculty of Geodesy, Geospatial and Civil Engineering, Institute of Geospatial Engineering and Real Estate, University of Warmia and Mazury in Olsztyn, Prawocheńskiego 15, 10-695 Olsztyn, Poland; cezary.kowalczyk@uwm.edu.pl (C.K.); jacek.kil@uwm.edu.pl (J.K.)

* Correspondence: krystyna.kurowska@uwm.edu.pl; Tel.: +48-89-523-42-81

**Abstract:** Road development projects are expansive and they exert a significant impact on the environment, landscape, spatial planning, and land management. In this study, we developed a research hypothesis that analysis of the shape factor of plots can be the basis for determining the factors affecting the level of urbanization. This article evaluates changes in the spatial structure of plots resulting from the construction of a city bypass. The proposed method is based on a morphological analysis of plots located in the vicinity of the motorway lane. In the next steps, lines located at a distance of 400 m and 800 m from the beltway were determined and then shape indicators were determined for the plots cut by these lines. The analysis confirmed the change in the shape of the plots, along with the distance from the beltway. Plots located further from the bypass of the city had smaller areas and the aspect ratio was similar for plots intended for development. The proposed method allows us to identify spatial effects occurring after entering a suburban road. The method should be used at the design stage of the beltway and not at the stage of impact assessment after its construction. This will allow for maintenance of a coherent spatial policy at the interface between urban and rural areas. At the same time, the study of changes in the morphology of plots allows earlier identification of urban processes.

**Keywords:** bypass; morphological structure; land plots; spatial order

## 1. Introduction

Cities are confronted with the pressing dilemma of how to improve the quality of life in the face of demographic changes, as follows: How to maintain, improve, and possibly increase the mobility of people and goods, while minimizing the negative consequences of transport, such as congestion, emissions, noise, loss of public space to traffic, and road safety [1]. The solutions to transport-related problems in urban areas have to be consistent with political priorities at the European, national, regional, and local level [1]. The absence of improvement in Eastern European roads could result from inappropriate policy design, which obstructs the development of Eastern European economies. The GDP of Hungary, Poland, Bulgaria, and Romania is still a small share of the EU's GDP, and income divergence contributes to uneven development [2,3]. The focus of planned transport policies should be on sustainable transport systems and on minimizing the adverse consequences of transport (congestion, air quality, noise, $CO_2$ emissions, and accidents) [4]. In an urban environment, the negative impact of automobilization can be counteracted by building bypasses. However, the advantages of bypasses are debatable in many respects [5]. In an era of rapid development, the main focus of research into urban transport has gradually shifted to sustainable planning and sustainable development methods [6,7]. According to many authors, the growth of the transport sector defies the principles of sustainable

development because it consumes natural assets and positions itself at the crossroads of economic, environmental [8,9], and social interests.

So far, research in Poland has focused mainly on the impact of motorway construction on the space, environment, agriculture in particular [10–13], the impact of the motorway on agricultural land [14], the spatial conflicts arising at the interface between the motorway and the natural environment [15], and the scope of effects in the vicinity of the motorway related to its construction and exploitation [16]. The direct impact of roads is usually analyzed within a radius of up to several hundred meters. However, linear referencing and the type of the construction project determine the extent of changes in the natural and cultural environment and the local landscape. The adverse consequences of motorway construction, such as spatial fragmentation (property rights and land use), unfavorable cropland distribution, and longer access roads are experienced within a 2 km radius from the planned roadway [16]. It mainly applies to agricultural land where intensive agricultural production is carried out.

In this time, more advanced research was conducted and in different aspects in the world, e.g., economic impact [17], business impact [18], environmental impact [19–21], impact on property values [22], accidents [23], etc. In evaluating bypass construction projects, rural areas (mostly those surrounding cities) should be analyzed from a somewhat different perspective. The changes resulting from the conversion of agricultural land to non-farming purposes, including services and housing, have to be taken into account. The construction of motorways (and bypasses) also necessitates changes in local zoning plans. The designation of areas zoned for various purposes could change and completely new functions could also be introduced in a given area [18].

Bypass roads and motorways exert a comparable influence on the surrounding areas due to similarities in their construction and operation. However, bypasses are usually built in rural areas due to strong urbanization pressure from the neighboring cities. Bypass roads occupy extensive stretches of farmland, which decreases the area of land under crops. Bacior [14] argued that the relevant risks should be evaluated based on analyses of changes in land use, soil quality class, and the distribution of access roads to land plots situated along the axis of the planned roads. Such analyses precede motorway construction projects, but their results are largely universal and can also be applied to city bypass roads. Other research has also identified different factors that exert a negative influence on the spatial structure of agricultural land, such as cutting off fields from the farmstead, and unfavorable size and distribution of plots [15]. Areas in the vicinity of motorways and changes in their spatial structure were also analyzed by Wilkowski [10]. His method can be used to evaluate motorway routes that intersect farmland to assess their influence on the surrounding areas and to select the optimal farm management strategies in rural areas affected by motorway construction.

The environmental impacts of motorway construction have been studied by Badora [15], and his findings can be extrapolated to roads. He proposed a methodology for a multi-criteria evaluation of a motorway's influence on the natural environment. The developed method combines indicators of environmental valuation with indicators of environmental transformation. Projects that induce extensive changes in farmland imply higher spending on the restoration of agricultural ecosystems. He also rightly noted that, due to the considerable degradation of agricultural ecosystems in the vicinity of the planned motorways, construction projects carried out in these areas make the greatest contribution to environmental protection [15].

Kozłowski [24] observed that the extent to which human activities influence the environment is largely determined by the natural value and the degradation of local ecosystems. The most serious conflicts can be expected in areas characterized by high natural value and significant degradation. Gwiaździńska-Goraj and Kurowska [25] analyzed two variants of the Olsztyn bypass. The analysis took into account environmental and social conditions. In their opinion, social expectations expose

valuable natural areas to degradation. Road planning requires extensive information campaigns to minimize the risk of social conflict, which substantially hinders and delays decision-making and construction. The relevant information has to be communicated not only to the owners of property directly covered by the roadway or in the direct vicinity of a planned road, but to all members of the local community. Local communities are mostly perturbed by the road's adverse impact on the local landscape and environment during construction and operation. Educational campaigns are needed to provide all stakeholders with reliable information about the extent and the environmental impacts of motorway projects as well as the planned preventive and repair measures.

The authors formulated the following research hypothesis: Analysis of the shape factor of plots can be the basis for determining one of the factors affecting the bypass of urbanized areas. While verifying the hypothesis, in this study a morphological analysis was carried out based on changes in land use, with particular emphasis on the structure of plots (area and shape) before and after bypass construction. A plot is understood as a part of the land area separated by fixed cadastral boundaries. The presented method is a new approach to assessing the effects of the location of road investments, in this case, a beltway.

Morphological analyses are frequently conducted in many fields of science, including geography, architecture, science, and philosophy [26–32]. This term is widely accepted and it applies to the morphology of cities [33–36]. This area of scientific inquiry is also referred to as urban morphology [37,38] or urban structure [32]. The study of urban morphology has fascinated numerous scholars since the formation of cities [39]. In Europe, urban morphology entered mainstream planning practice in the twentieth century [31]. A morphological analysis of spatial data is a new element of research into sustainable transport planning and spatial order. Spatial order is evaluated in land management analyses, it plays an important role in local development, and it is an object of scientific inquiry in geodesy and cartography, mainly in relation to cities and spatial development. Spatial order is increasingly regarded as an integral element of the sustainable development concept [40–43].

As part of the conducted research, the proposed method is based on a morphological analysis of plots located in the vicinity of the motorway lane. In the next steps, lines located at a distance of 400 m and 800 m from the beltway were determined and then, for the plots cut by these lines, shape indexes and surface parameters were determined. The analysis allowed us to conduct a discussion and formulate conclusions. In this study, GIS tools were used in morphological analysis. The application of GIS tools increases the reliability of spatial analyses, thereby minimizing the negative impact of transport networks on the environment and the spatial structure of neighboring areas and reducing social conflicts that surround the construction of new roads.

## 2. Materials and Methods

### 2.1. Study Area

The morphological features of the areas directly adjacent to the Olsztyn bypass were analyzed. The study was performed on the Olsztyn bypass with an estimated length of 28 km (including the bypass road of Wójtowo). The analyzed bypass intersects the municipalities of Gietrzwałd, Stawiguda, Purda, and Barczewo as well as Olsztyn, the capital city of the Region (Voivodeship) of Warmia–Mazury, which has county status (Figure 1). The bypass road stretches from the village of Kudypy, west of Olsztyn, to the village of Wójtowo, east of Olsztyn, and it is connected to the national road No. 16 at both ends. It consists of two roadways with two lanes in each direction. The Olsztyn bypass will ultimately have six road junctions, 36 engineering structures, and 32 culverts [44].

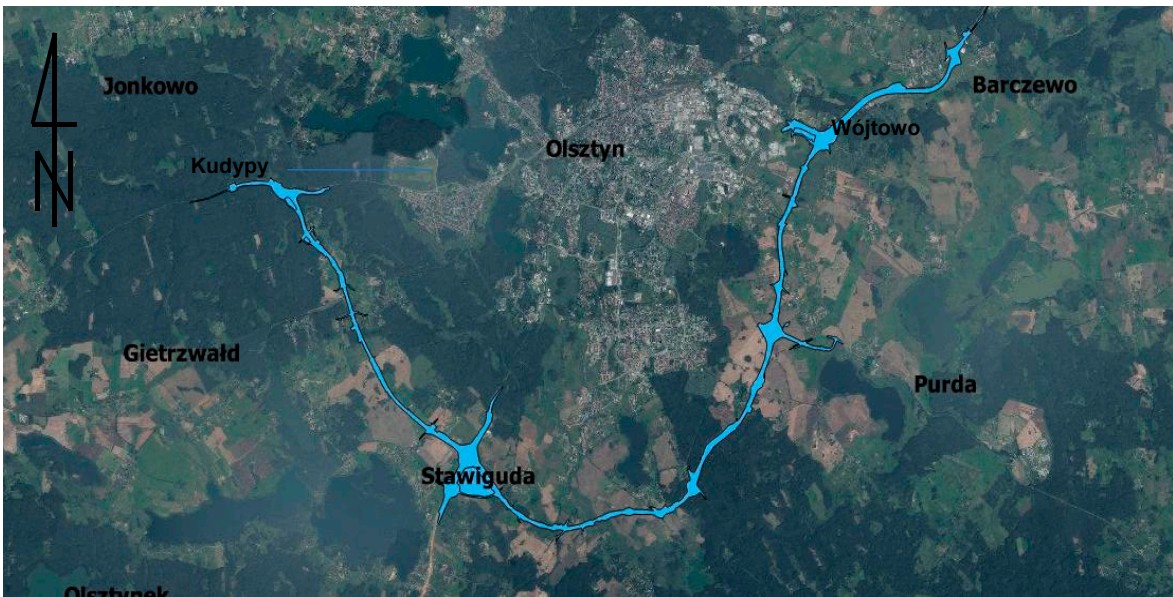

**Figure 1.** Part of the Olsztyn with localization of the bypass (the beltway is marked in blue). Scale 1:168,000.

The Olsztyn bypass intersects mainly agricultural areas situated far from dense development and village centers. Short sections intersect forests, clusters of field trees, and scattered settlements. In three locations, the bypass crosses railway lines connecting Olsztyn with the neighboring towns. It also intersects the Łyna River and the Szczęsne Canal. The bypass road will also intersect nature conservation areas and the surrounding territories, including the following:

- Protected Landscape Area of the Pasłęka Valley;
- Protected Landscape Area of the Middle Łyna Valley;
- Protected Landscape Area of the Napiwodzko-Ramucka Primeval Forest [45];
- Special Bird Protection Area of the Napiwodzko-Ramucka Primeval Forest, which is part of the Natura 2000 network.

### 2.2. Method of Analysis

In this study, the analysis focused solely on changes in morphological features resulting from the division or consolidation of land. The cadastral land plot is the smallest unit of geodetic division in Poland and it was adopted as the basic unit of analysis in this study. Cadastral plots can be divided or consolidated, which leads to changes in their spatial structure.

Thousands of cadastral plots have to be divided in the process of planning and designing a bypass road. The physical boundaries of a roadway denote the extent to which land plots are occupied and split by the motorway. Linear referencing in physical space induces significant changes in the natural environment and creates new spatial structures in agricultural areas. The route of a bypass road significantly alters local morphological features. In this study, only the morphological features of the evaluated land plots (separated by cadastral borders) were analyzed, i.e., located at a maximum distance of 800 m from the road axis. The maximum distance was determined based on the extension of plots on the test area (the farthest line intersects the plot whose shape has not been changed while determining the route, as seen in Figure 2, level I).

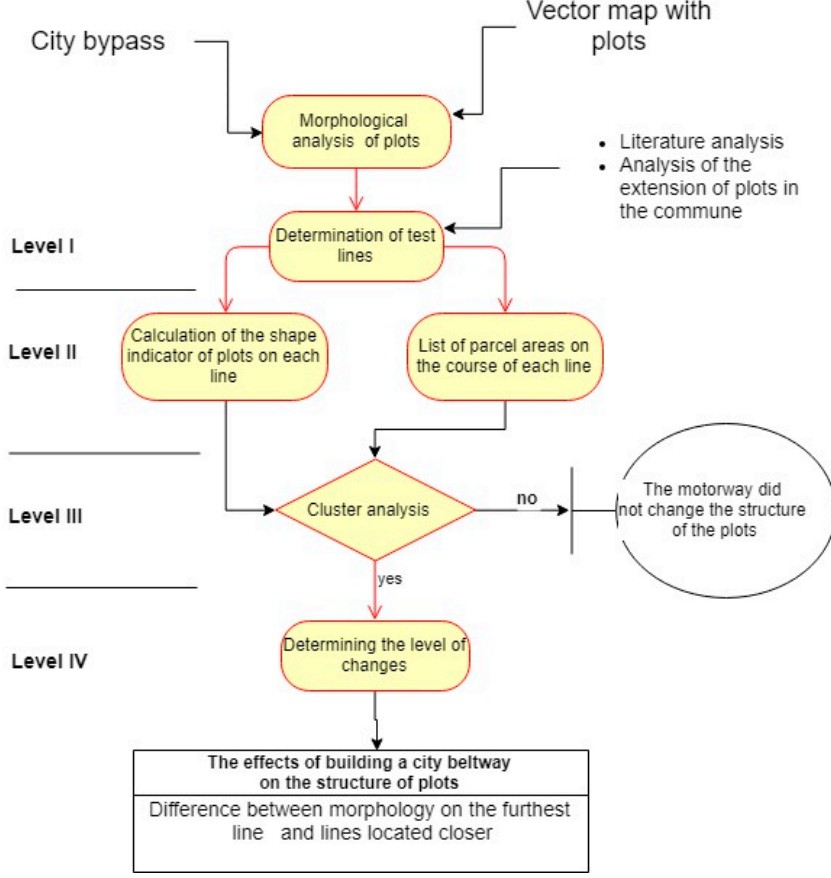

**Figure 2.** Flow chart of the analysis.

Variations in the land plot mosaic were evaluated on the assumption that the perimeter and the area of land plots have a structure adopted in the research as the starting point (800 m). The bypass road will change the above parameters and the extent of these changes will be determined by a land plot's distance from the bypass. The above distances were selected to evaluate the land plots adjacent to the bypass on the assumption that the contribution of the intersected land plots to the calculations of the neighboring line will not exceed 5% (ratio of the number of land plots repeated on the neighboring line to the number of plots intersected by a given line). The influence of the intersection on the primary structure (morphology) decreases with an increase in distance from the bypass [46]. The adoption of 5% of plots is necessary due to the fact that among plots of land through the city beltway are also plots of roads (plots of roads are cut by all lines).

Level II of the analysis was the determination of shape indicators and plot area on individual test lines. The shape factor was adopted as the basic indicator of an area's morphology. Analyses of two-dimensional objects are often used in settlement geography. There is a lot of indexes that can give supportive information regarding the shape metric [47]. The shape factor was determined based on the object's basic parameters, surface area and perimeter. Indicator k (Equation (1)) was proposed by Kostrubiec [48]. This shape indicator relies on a land plot's area (A) and perimeter (P) and is calculated with the use of the following formula:

$$k = (P^{2/}A) - 4 \times \pi. \tag{1}$$

Shape factor k is a measure of an object's "cohesion". It adopts a minimum value equal to zero for a circle and it increases as the object becomes more elongated. Shape factor k takes on an infinite value for an infinitely narrow rectangle [30]. The shape factor equals 5.44 for a rectangle, with an aspect ratio of 1:2, and 3.44 for a square.

Level III of the analysis was the determination of the difference of plots located near the beltway in relation to plots that have not been cut by the bypass, i.e., plots on the farthest test line.

As the first stage of analysis, we performed similarity analysis (cluster analysis). This analysis allowed us to confirm or deny the occurrence of differences in the morphology of plots situated at different distances from the beltway. If the plots on particular lines are grouped with the same distance, then one can proceed to the next level of analysis (level IV). Otherwise, there is a basis for stating that there is no change in the morphology of plots.

Level IV (Determining the level of changes) analysis was performed when cluster analysis showed grouping within plots on lines located at the same distance. The variations in shape $\gamma$ were calculated (Equation (2)) based on the difference in the mean shape factor of plots intersected by line a3 ($k_{av\_an}$—average shape factor of plots located on the line located further) and line a2 ($k_{av\_a(n-1)}$—average shape factor of plots located on a line closer to one another), relative to the shape factor of plots intersected by line a3 ($k_{av\_an}$).

$$\gamma = (k_{av\_an} - kav_{\_a(n-1)})/k_{av\_an}. \tag{2}$$

The changes in plot area $\delta$ were also calculated (Equation (3)) based on the difference between the average area of land plots intersected by line a3 ($A_{av\_an}$—average area of plots located on the line located further) and line a2 ($A_{av\_a(n-1)}$—average area of plots located on a line closer to one another), relative to the area of plots intersected by line a3.

$$\delta = (A_{av\_an} - A_{av\_a(n-1)})/A_{av\_an}. \tag{3}$$

### 2.3. Input Data

The analysis was performed on cadastral land plots in the vicinity of the Olsztyn bypass. The municipalities intersected by the bypass are presented in Table 1. They had a similar average plot, which ranged from 16,442 in the municipality of Barczewo to 25,326 in the municipality of Purda.

**Table 1.** Characteristic features of the municipalities intersected by the Olsztyn bypass (data from 2018).

| Municipality | Total Area (km$^2$) | Population (Persons) | Number of Plots | Average Plot Area (m$^2$) | Average Plot Perimeter (m) |
|---|---|---|---|---|---|
| Stawiguda | 222.9 | 8449 | 10,886 | 20,454 | 447 |
| Barczewo | 319.85 | 17,662 | 19,446 | 16,442 | 463 |
| Purda | 318.1 | 8612 | 12,544 | 25,326 | 557 |
| Gietrzwałd | 172.3 | 6536 | 9565 | 17,993 | 470 |

The number of land plots was similar in the rural areas of the analyzed municipalities. Barczewo was characterized by the largest average plot area of 1.6 ha. The shape of a land plot was determined by its surface area and perimeter. The examined plots had the shape of narrow rectangles. Materials for analysis (plot boundaries recorded in vector form) were obtained from the Geodetic and Cartographic Documentation Centre in Olsztyn. The data validity period was 2018.

The route of the Olsztyn bypass against the land plot mosaic is presented in Figure 3. With respect to the road axis, two research lines were constructed. The first was 400 m from the road and the second was 800 m from the road. Based on the literature analysis [14,15], the distance of 400 m was selected as the distance of the ring road direct impacting the environment (parcels crossed by a line located at a distance of 400 m were partially crossed by the separated road), while 800 m was the distance accepted by the authors for the purposes of the study as the range of parcels that were not crossed by a road lane. When it came to areas with a different use structure (larger areas of parcels, typically agricultural use), the research lines were adapted to the use structure.

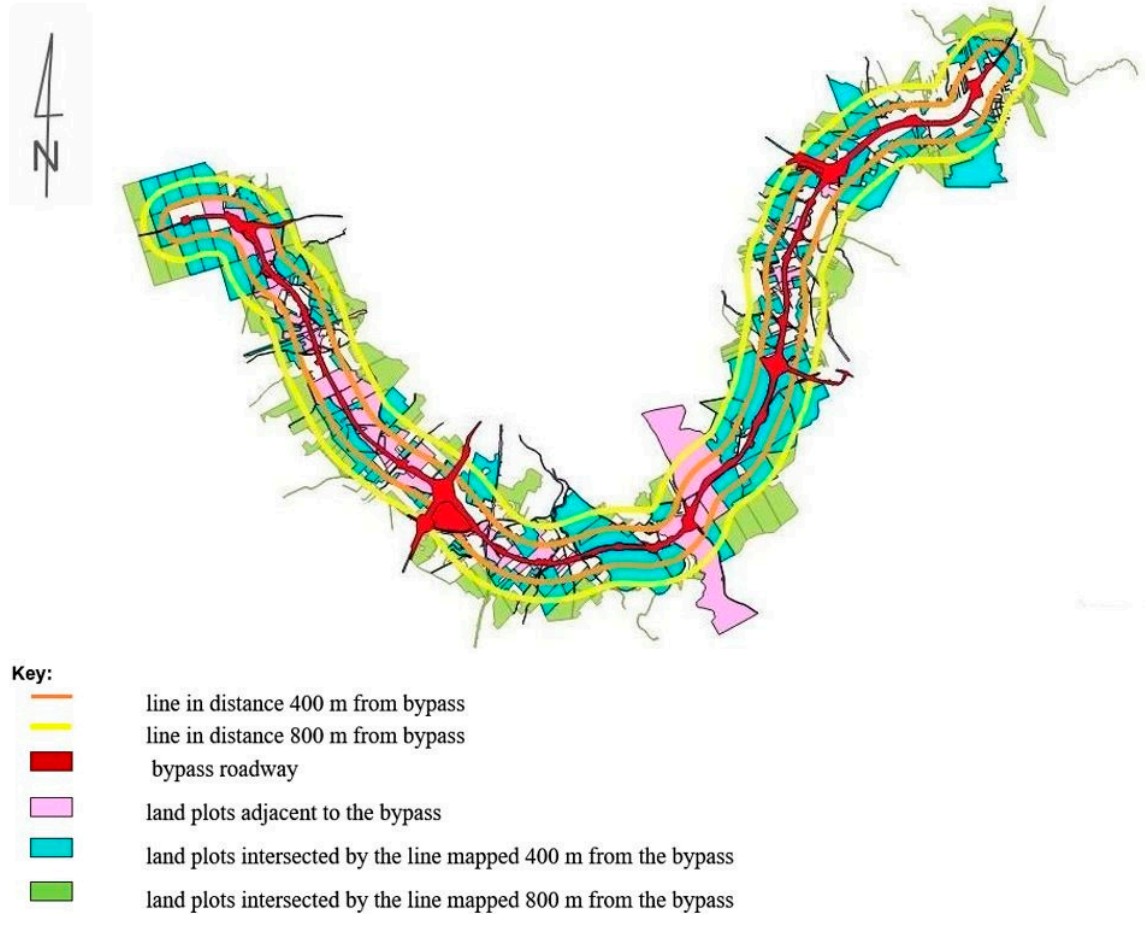

**Figure 3.** The route of the Olsztyn bypass with an indication of the analyzed land plots.

## 3. Results

The study was performed along a 28 km section of the Olsztyn bypass, including cadastral plots directly adjacent to the bypass road on both sides. GIS software [49] was used to map two lines parallel to the road axis, line a2 at a distance of 400 m from the road axis and line a3 at a distance of 800 m from the road axis. The road axis was marked by line a1 (Figure 3). The spatial correlations between the above lines were analyzed. Datasets describing every land plot, including land plot number, perimeter, area, and orientation (N—north and S—south), intersected by each line were exported. Numeric maps were analyzed to calculate the shape factor of every land plot and the mean values of shape factors. The results are presented in Table 2.

**Table 2.** Descriptive statistics of shape factor of plots intersected by lines at various distances from the bypass.

| Line | Orientation | Number | Shape Factor | | | | |
|------|-------------|--------|---------|--------|---------|---------|--------|
| | | | Average | Median | Minimum | Maximum | SD |
| a1 | N | 86 | 208.55 | 30.29 | 3.92 | 1217.72 | 305.06 |
| | S | 96 | 175.28 | 40.6 | 3.92 | 1217.72 | 274.25 |
| a2 | N | 307 | 78.73 | 78.73 | 2.66 | 2438.34 | 212.99 |
| | S | 209 | 86.3 | 12.1 | 2.99 | 1276.95 | 191.48 |
| a3 | N | 238 | 63.92 | 9.59 | 2.47 | 954.97 | 148.99 |
| | S | 294 | 92.53 | 10.79 | 2.55 | 4121.37 | 292.97 |

The presented results indicate that land plots directly adjacent to the bypass roadway had higher shape factors on average, which implied that they were more elongated (Table 2 and Figure 3), with elongation indicated as the ratio of width to extension. Elongation is the average aspect ratio determined on the basis of the shape index.

Land plots on the northern side of the bypass were more elongated than those situated on the southern site, and their aspect ratios were determined at 1:35 (N) and 1:45 (S), respectively. Land plots directly adjacent to line a1 (road axis) were not highly suitable for agricultural production. Land plots intersected by lines a2 and a3 were characterized by more desirable proportions. Plots situated 400 m away from the roadway had an average shape factor of 78.73 (northern side) and 86.3 (southern side), with an aspect ratio of 1:21 (northern side) and 1:23 (southern side), respectively. Similar observations were made in land plots situated at a distance of 800 m from the roadway and intersected by line a3. These plots had an average shape factor of 63.93 (northern side) and 92.53 (southern side), with an aspect ratio of 1:17 (northern side) and 1:24 (southern side), respectively.

The analysis of the similarity of the parcel shapes on the course of individual lines was carried out on the basis of clustered analysis (agglomeration method, Figure 4). The agglomeration method that was used when forming clusters based on the distances between objects was applied. Clusters of objects (the shape factor of every land plot) were determined based on Euclidean distances, i.e., the geometric distance within a multi-dimensional space.

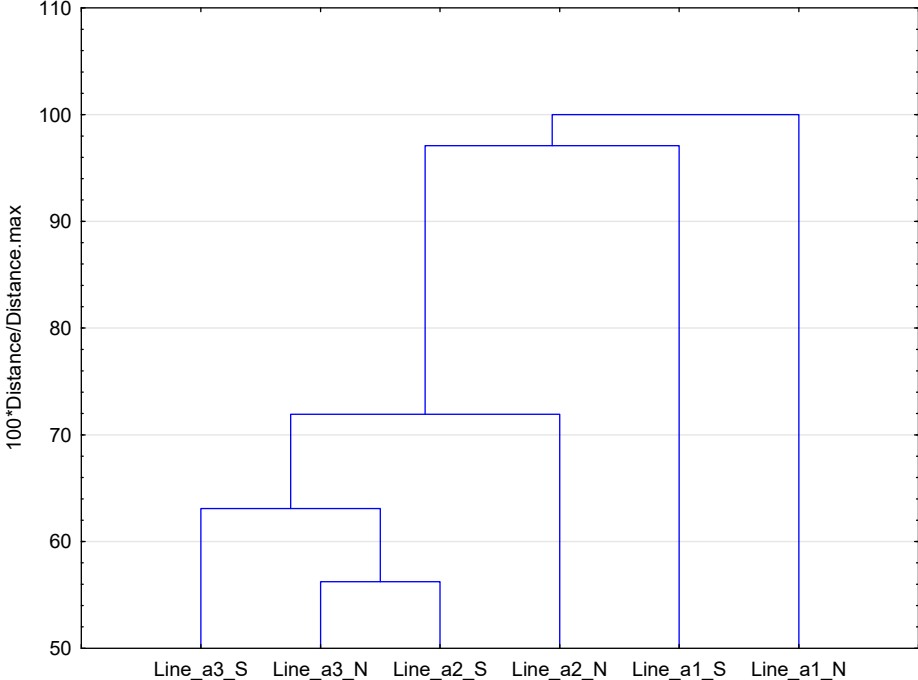

**Figure 4.** The analysis of similarity of the parcel shapes on the course of individual lines (clustered analysis).

Analysis of similarity allowed for the grouping of lines, which were located at different distances, by grouping the most similar plots according to shape. The first group of plots with similar shapes was on the lines a3_N and a2_S. The plots located on the a3_S line were very similar to the indicated group. Plots located near the beltway (plots on the lines a1_S and a1_N) showed the least similarity to the group of plots on lines a3_N and a2_S.

Land plots directly adjacent to the bypass were characterized by less desirable proportions than those situated further away from the roadway. This can probably be attributed to the suburban character of the evaluated area and a large number of roads that were divided by the bypass. The plots occupied by roads were long and narrow, which explains the average values of their shape factors and aspect ratios.

The shape factors were presented separately on both sides of the bypass in a scatter plot (Figures 5 and 6) to illustrate the changes in the morphological properties of the analyzed area.

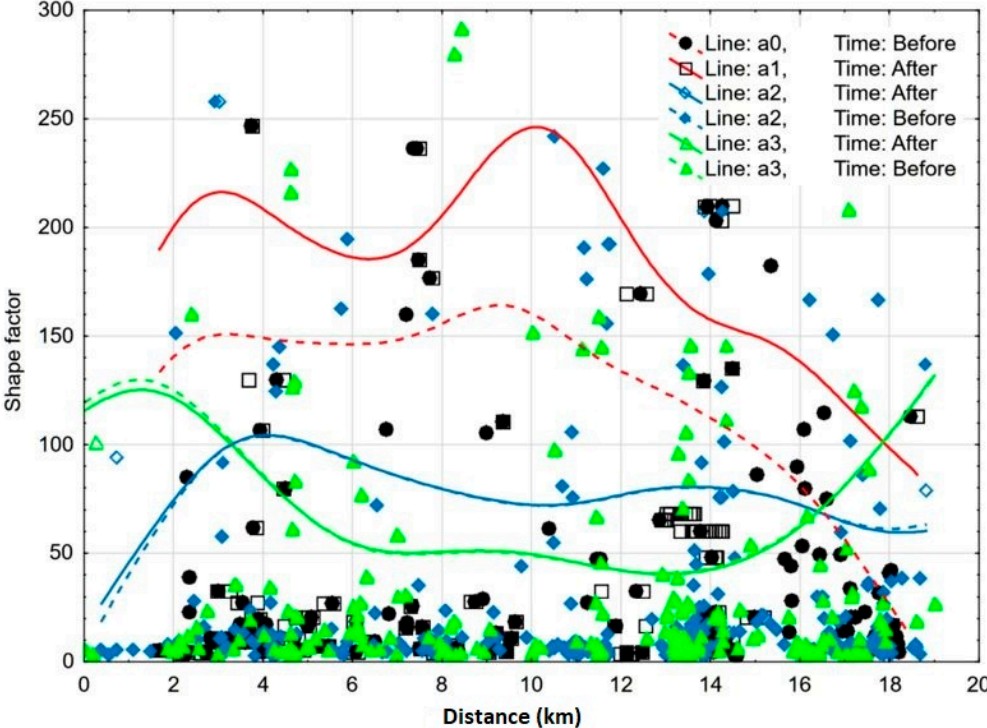

**Figure 5.** Scatter plot of shape factor values for land plots on the northern side of the bypass (the lines were smoothed with the distance-weighted least squares procedure).

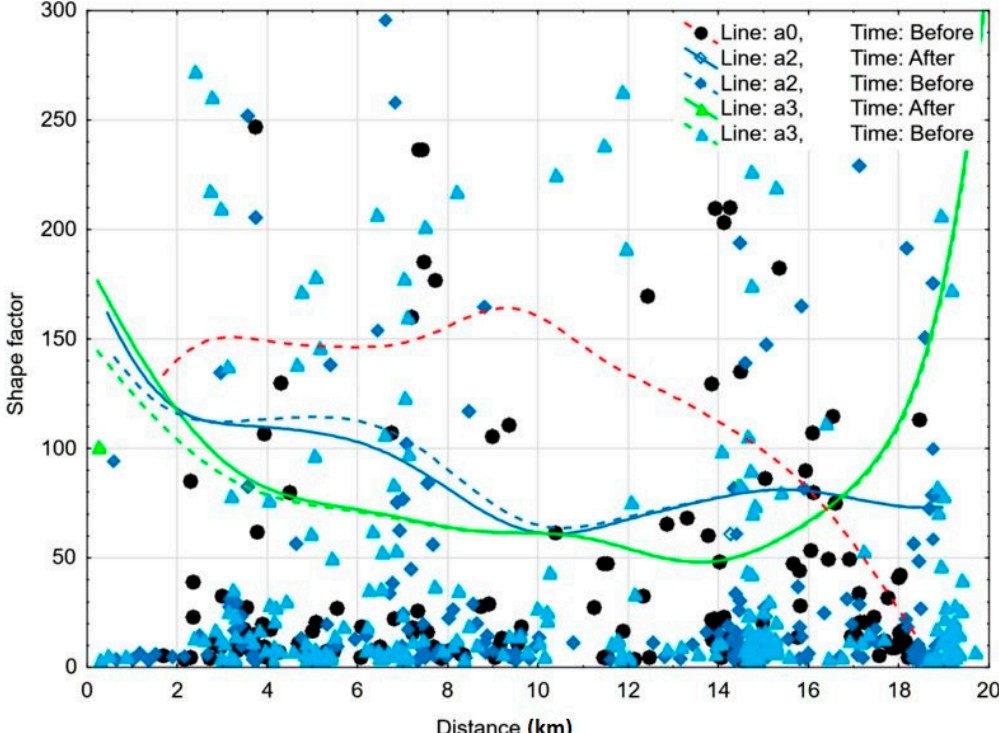

**Figure 6.** Scatter plot of shape factor values for land plots on the southern side of the bypass (the lines were smoothed with the distance-weighted least squares procedure).

The areas of land plots situated at various distances from the bypass are presented in Table 3. The average area of plots intersected by line a1 was 12.5 ha on the northern side and 14.4 ha on the southern side. The average area of plots intersected by line a3 was 2.9 ha on the northern side and 3.3 ha on the southern side.

**Table 3.** Descriptive statistics of the area of plots intersected by lines at various distances from the bypass.

| Line | Orientation | Number of Plots | Plot Area (m$^2$) | | | | |
|------|-------------|-----------------|---------|--------|---------|---------|----|
| | | | Average | Median | Minimum | Maximum | SD |
| a1 | N | 86 | 125,867.48 | 46,986.62 | 353.84 | 1,763,222.20 | 263,624.61 |
| | S | 96 | 144,172.84 | 66,361.43 | 324.00 | 1,763,222.20 | 255,480.65 |
| a2 | N | 307 | 38,422.92 | 4090.36 | 153.16 | 963,945.02 | 105,522.81 |
| | S | 209 | 61,858.48 | 8263.47 | 202.21 | 1,095,084.56 | 144,933.35 |
| a3 | N | 238 | 29,130.72 | 3197.07 | 105.3 | 405,716.90 | 68,058.58 |
| | S | 294 | 32,706.19 | 4773.69 | 145.94 | 426,111.94 | 73,217.24 |

As shown in Figure 4, the shapes of the plots changed depending on the distance from the beltway. In order to check whether the distribution of parcel areas also changed depending on the distance from the beltway, a similarity analysis was carried out. Analysis of similarity allowed the grouping of lines that were located at different distances into groups with the most similar parcel surfaces (Figure 7). The first group of parcels with similar shapes was on lines a3_N and a3_S. The plots located on the a2_S line were very similar to the indicated group. Plots located near the beltway (plots on lines a1_S and a1_N) showed the least similarity to the group of plots on lines a3_N and a3_S. Generally, it could be concluded that plots located at a greater distance from the road line had smaller areas.

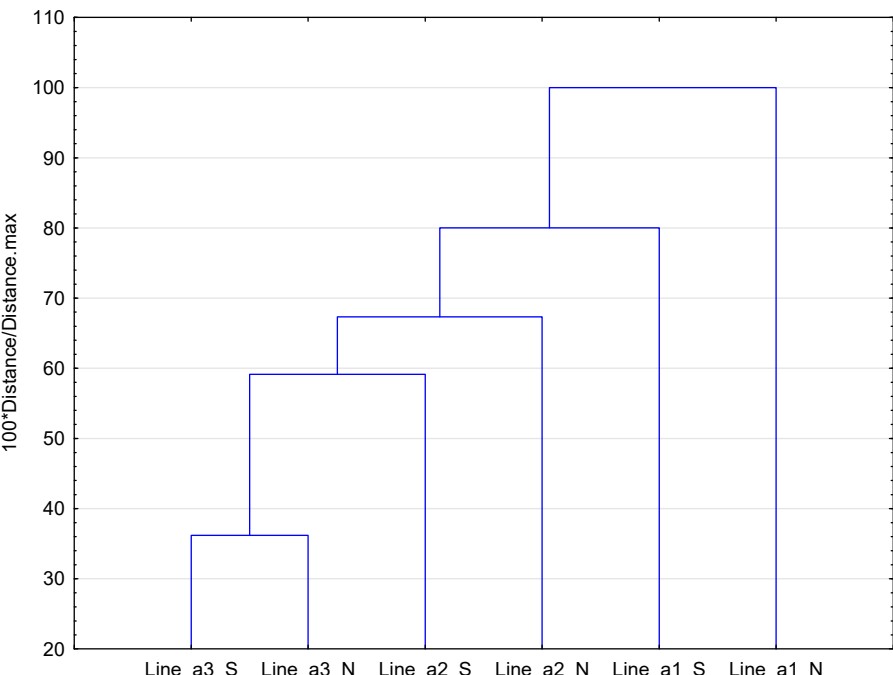

**Figure 7.** The analysis of similarity of parcel surfaces on the course of individual lines (clustered analysis).

The distribution of land plots intersected by the above lines was presented in scatter plots, as seen in Figures 8 and 9. The x-axis represents the distance from the start point and the y-axis represents plot area in m$^2$. The lines smoothed by the distance-weighted least squares method procedure were

fitted into the model. The results indicate that smaller plots were situated further from the bypass. As previously noted, the Olsztyn bypass intersects agricultural land in the suburban zone. It bypasses clusters of residential development and line a3 (800 m from the bypass) intersects land plots zoned for the construction of single-family homes, whose area is relatively small in comparison with the agricultural plots adjacent to the bypass.

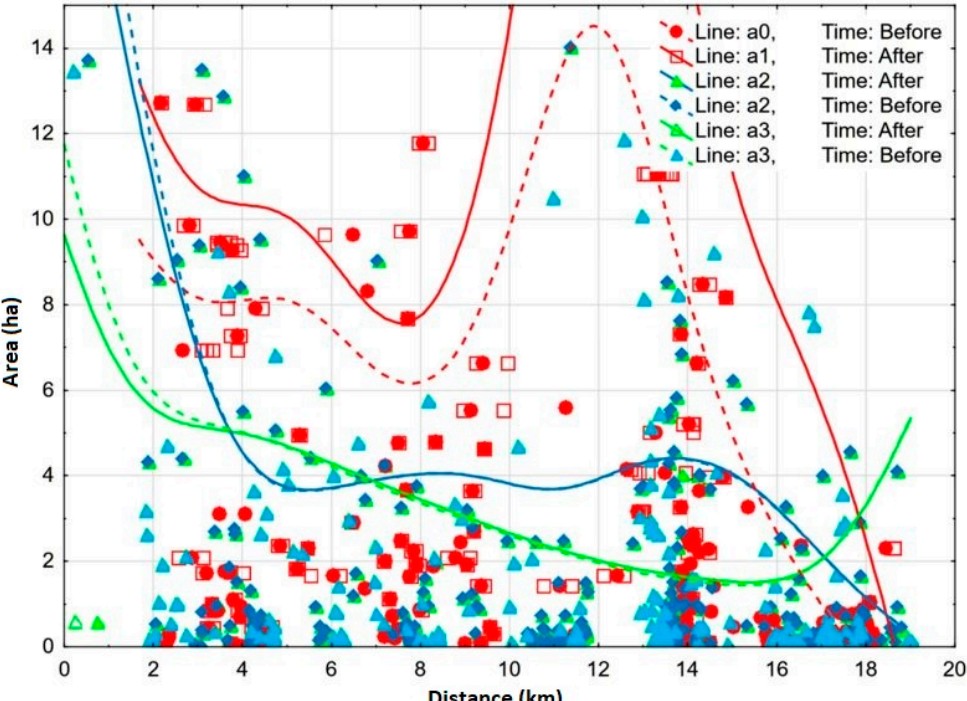

**Figure 8.** Scatter plot of plot areas on the northern side of the bypass (the lines were smoothed with the distance-weighted least squares procedure).

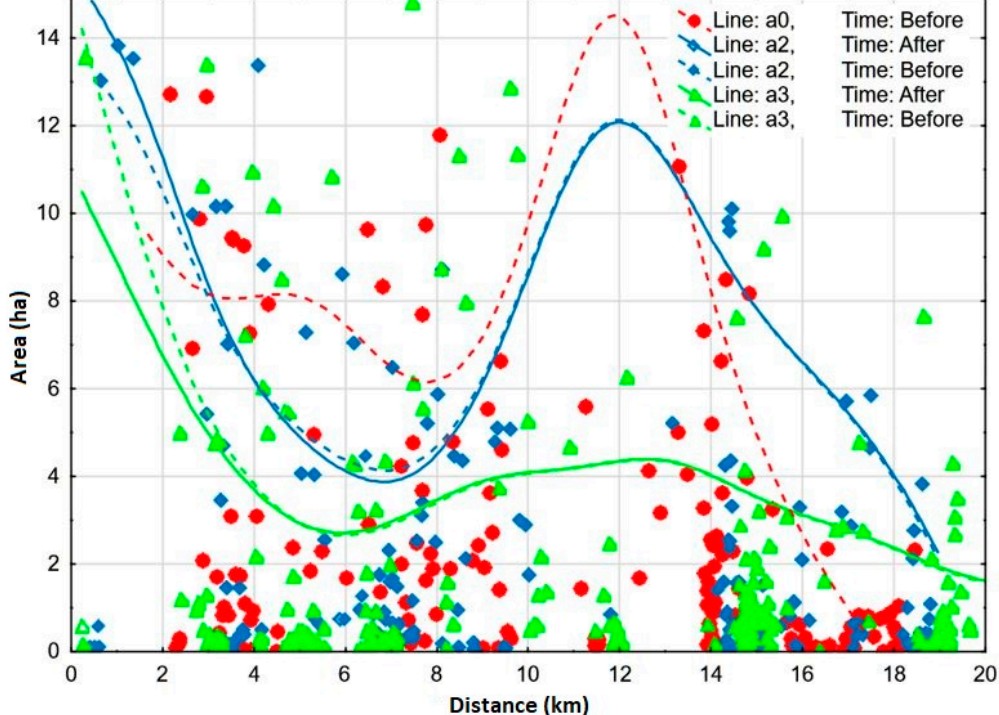

**Figure 9.** Scatter plot of plot areas on the southern side of the bypass (the lines were smoothed with the distance-weighted least squares procedure).

The extent of changes in shape factor values were determined, as seen in Table 4. Land plots situated closer to the bypass road had a higher shape factor (the mean shape factors of land plots intersected by extreme lines differed by up to 220%).

**Table 4.** Changes in the shape factor of land plots intersected by the mapped lines.

| Change between Lines | Orientation | Change in Shape γ |
|:---:|:---:|:---:|
| a3–a1 | N | −226% |
|  | S | −89% |
| a2–a1 | N | −23% |
|  | S | 7% |
| a3–a2 | N | −165% |
|  | S | −103% |

The greatest differences in average plot area were determined in the plots that were intersected by line a1 and were directly adjacent to the bypass on both the northern and southern side. The results are presented in Table 5.

**Table 5.** Changes in the average area of land plots intersected by the mapped lines.

| Change between Lines | Orientation | Change in Plot Area δ |
|:---:|:---:|:---:|
| a3–a1 | N | −332% |
|  | S | −341% |
| a2–a1 | N | −32% |
|  | S | −89% |
| a3–a2 | N | −228% |
|  | S | −133% |

## 4. Discussion

The interaction between accessibility and land use has attracted considerable attention. Land use is a significant aspect of accessibility measurement and also the key to urban environmental design [50]. Similar to motorways, bypasses decrease plot area, increase the number of land plots, and increase the distance between the farmstead and fields, in the case of agricultural farms [18]. A simplified approach to evaluating the severity of functional changes in farmland was proposed by Chmielowiec and Kaszycki [51]. The authors divided farms into groups based on the percentage of farm area affected by functional changes.

The above data were analyzed to identify changes in land parameters that influence agricultural productivity, and the results were used to evaluate a motorway's impact on cropland. The methods used support a comprehensive assessment of a motorway's influence on farmland [17,52], including the loss of land reserves for motorway construction, decreases in the productivity of farmland situated in the vicinity of the motorway, and fragmentation of cropland intersected by the motorway [14]. The use of agricultural land is considered as an intermediate human impact on nature and urban ecosystems [53,54]. The conversion of agricultural land to urban or suburban areas is the process which usually has a place around cities. It can further degrade the provision of beneficial ecosystem services [54,55]. New road projects also exert a significant influence on the creation of land reserves. The impact of road construction projects is multidimensional and, although it directly affects land reserves, it also indirectly leads to a decrease in farmers' annual incomes and significantly increases public expenditures related to the purchase of the remaining land [56].

Previous studies have confirmed that transport planning affects land use and that land use influences transport accessibility, which is the ease of travel activity and of reaching destinations [54,57,58].

Changes in accessibility and transportation infrastructure are likely to influence the attractiveness of a location, land use, human activities, and property value, whereas the converse impacts of transportation infrastructure are less understood [50,59].

A dense and suitably equipped road network promotes fast and convenient transport and is essential for the economic growth of every country. It also creates opportunities for eliminating economic exclusion zones in the poorest regions [46]. The study by Zhou et al. offers highly insightful observations [38]. The authors explored the relationship between urban growth and urban accessibility, in an anatomic spatial-temporal fashion, in view of the number and size of land use parcels developed over time, urban accessibility from residential to non-residential land use areas, and the statistical relationships between urban form and urban accessibility [38].

The demand for residential land use is relatively stable and responsive to population growth, while the land demand for other social, economic, or technological needs is less responsive to population growth. Commercial, office, or industrial land uses, such as shopping malls, office complexes, and industrial parks, are often characterized by cyclical and varied development in the longer term [60].

Tong et al. [50,61] found that the location of major roads also has a noticeable effect on changes in land use. On the other hand, due to the location of the beltway, we can observe an intensified urban sprawl [62–64]. The shape and size of the plots created after the construction of a bypass considerably influence the direction of development of these areas. Bypass construction also necessitates changes in local zoning plans [65]. The designation of areas zoned for various purposes could change and completely new functions could also be introduced in a given area. Therefore, the cycles in the relationship between form and the investment that generated it can be observed by studying changes in urban form at the level of land subdivisions. Regeneration sets out to enable a city or neighborhood to attain a self-perpetuating state of complexity and diversity beyond the capacity of market forces [36].

According to Elias and Shiftan [66], it is generally assumed that bypass construction exerts a negative impact on the local economy by reducing traffic for businesses. In contrast, bypass construction improves access to local communities, promotes the development of residential and commercial facilities, and increases economic activity [66]. The locational shift in traffic can force the existing businesses to close up or relocate, but it can also create new business opportunities. In the longer term, bypasses can exert profound effects on development patterns, but in smaller cities, their impact can be felt only after 20 or more years [67].

In our research, the land plots with the smallest average area were situated furthest from the Olsztyn bypass (800 m). The above indicates that land plots that are not directly adjacent to the roadway (line a3) are more attractive, due to easy access to the city. For this reason, larger plots are divided into smaller plots for residential construction, especially in the vicinity of villages. Land plots directly adjacent to the bypass are less attractive for urban development. In the rural municipalities surrounding the urban core, suburbanization begins with residential construction, and areas directly adjacent to expressways are not attractive for this purpose.

Land plots situated further from the bypass have more desirable shape factors, which were determined to be 63.93 (northern side) and 92.53 (southern side) for the plots intersected by line a3 (800 m), which have an aspect ratio of 1:17 (northern side) and 1:24 (southern side). Land plots intersected by line a1 and adjacent to the bypass have an aspect ratio of 1:53 (northern side) and 1:45 (southern side). The average aspect ratio of land plots intersected by line a1 can be attributed to the fact that most of them are long and narrow and are utilized as roads. Bypass construction disrupted the spatial structure of the analyzed plots and produced irregularly shaped remnants of larger parcels, which also influenced the observed results. A dense road network in the analyzed area is probably responsible for the elongated shape of the examined plots.

The advantage of the proposed method is the rapid identification of changes in space (changes to the morphology of plots), which allows identification of urban planning processes. The existing methods of analysis of changes consisted of studying factors reflecting the number of buildings.

Some disadvantage of the method may be accessing a large number of vector data. However, in the era of today's technical possibilities and cadastral public resources, this is likely a minor disadvantage.

## 5. Conclusions

The morphology of the spatial structures modified by the construction of the Olsztyn bypass was analyzed to determine the following:

- The correlations between changes in plot area and the location of the Olsztyn bypass; and
- Plot shape characteristics of the analyzed area.

The examined suburban area is difficult to analyze because it constitutes a transitional zone between the urban area of Olsztyn and the rural areas of the surrounding municipalities. It is characterized by overlapping areas of clustered and dispersed single-family homes and farmland. The analyzed area also features extensive fragments of two forest complexes and a road network.

This study analyzed a relatively short fragment of the Olsztyn bypass. Therefore, it does not support the formulation of general conclusions regarding changes in spatial morphology, particularly in transitional suburban zones (at the contact point of urban and rural areas).

The presented method allows one to determine whether there was a disturbance of the morphology of plots due to the location of the beltway. If so, the level of change in morphology can be estimated.

As other studies show [30], urbanization processes are faster in areas where plots are smaller. The proposed method should be used at the design stage of the beltway and not at the stage of impact assessment after its construction. This will allow for maintenance of a coherent spatial policy at the interface between urban and rural areas.

Further research is required to validate the results of this study in other areas where bypass roads are planned. An additional aspect to be tested in subsequent stages is to determine the rate of change in the morphology of plots after the construction of the beltway, which will allow us to determine the rate of changes in the suburban area.

**Author Contributions:** Conceptualization, C.K. and J.K.; methodology, C.K.; formal analysis, K.K.; investigation, J.K.; data curation, J.K.; writing—original draft preparation, K.K. and C.K.; writing—review and editing, K.K.; visualization, C.K. and J.K.; supervision, C.K.; project administration, C.K.

**Funding:** This research received no external funding.

**Conflicts of Interest:** The authors declare no conflict of interest.

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
