# Peer review of "Changes in Land Plot Morphology Resulting from the Construction of a Bypass: The Example of a Polish City"

_sustainability, doi:10.3390/su11102987_

Round 1
Reviewer 1 Report
The article is very well completed. The rich literature and statistical studies have raised the values of the article. Accept in present form.
Author Response
Thank you very much for acceptation if uur manuscript.
Sincerely,
Autors
Reviewer 2 Report
The manuscript had some improvements since the last submission. However I suggest again some changes.
Although I'm not fully convinced about your approach, you should discuss the pros and cons of your method in discussion section.
The conclusion section seems more similar to a discussion of your results. I suggest to move this part to discussion and restructure the Conclusions section (also adding for example the future research directions).
in Fig.7 you provided a cluster analysis. Please, add more details about similarity rules used, reasons why you used such analysis and expected results.
Author Response
Manuscript ID: sustainability-491060
Title: Changes in Land Plot Morphology Resulting from the Construction of a Bypass on the Example of a Polish City
Dear Reviewer,
Thank you very much for your comments and suggestions which we have utilized to improve the quality of the paper. We believe it is ready for publication.
According to the reviewers' suggestions, the English language was corrected by the editors from MDPI
Authors

Reviewer 3 Report
A well-written article, with huge potential in sustainable development. There are some minor notes on the paper followed by a major comment which I think can improve the paper significantly:
Minor issues:
Abstract: can be improved by discussing the impacts of the paper and how it can help various stakeholders of such a development.
Introduction, Line 70, Page 2: “In other research has … “ please delete ‘In’
Material and method: Line 204, Page 6, there is a crossed line: “The construction of the bypass”
Discussion Line 356, Page 13, reference is needed for the sentence “… by Chmielowiec and Kaszycki”
Suggest mentioning the application of automation and machine learning for handing big data in the discussion.
Line 398, Page 14, “According to Elias and Shiftan..” keep the reference number, i.e. [65] by the authors name.
Major comment:
The application of the developed method is not very clear; therefore, I suggest adding an alternative route for the bypass to compare the results with the existing analysis. Then you could be able to say that the developed method has an application in selecting the most sustainable route for a road when considering land plot morphology.
Author Response

(The authors gave the same response as above.)

Reviewer 4 Report
The paper "Changes in Land Plot Morphology Resulting from the Construction of a Bypass on the Example of a Polish City" is interesting and original, within the scope of Sustainability. I think the content is innovative and never published. The review of the state-of-the-art is well structured and organized. The study is well organized. I have some questions for the authors:
please review the English language;
in Figure 1 add the scale and omit some city names that are not readable. Please add the N symbol;
Figure 2 could be improved;
cite with the number all Equations before they appear;
please revise lines 204-205;
in Table 1 revise the unit of measure of the area of measure (the superscript of m2).
Author Response
Manuscript ID: sustainability-491060
Title: Changes in Land Plot Morphology Resulting from the Construction of a Bypass on the Example of a Polish City
Dear Reviewer,
Thank you very much for your comments and suggestions which we have utilized to improve the quality of the paper. We believe it is ready for publication.
According to the reviewers' suggestions, the English language was corrected by the editors from MDPI
Sincerely,
Authors
Round 2
Reviewer 2 Report
The manuscript reached an acceptable level and it is now suitable for publication in Sustainability journal.
I have only one comment:
lines 365-366: the authors wrote "The conversion of agricultural land to urban or suburban areas is an unavoidable process." I think this sencence is a bit too strong. The urbanization process usually occurs on agricultural land, but not all agricultural areas undergo transition on urban areas.
Author Response
Manuscript ID: sustainability-491060
Title: Changes in Land Plot Morphology Resulting from the Construction of a Bypass: The Example of a Polish City
Dear Reviewer,
Thank you very much for your comments and suggestions which we have utilized to improve the quality of the paper. We believe it is ready for publication.
Authors
This manuscript is a resubmission of an earlier submission. The following is a list of the peer review reports and author responses from that submission.
Round 1
Reviewer 1 Report
Abstract:
1. There is no information about the aim of this study.
2. There is no information about the the results of research
Introduction:
3. Lines from 104 to 110 should be moved to the discussion
4. At the end of the introduction, there is no information about research hypothesis (lines 121-130)
Materials and Methods:
5. There is a detailed description of study area, but please indicate graphically which part of the beltway has been analyzed. Can you show map or drawing?
6. There is no information whether the analyzed section of the beltway runs through the municipalities given in the table 1
7. There is no short and clear information about the research methodology (we can find detailed desciption)
8. Lines from 188 to 210 should be moved to the results
Discussion
9. Lines from 298to 315 should be moved to the introduction or removed
Author Response
Thank you for your valuable comments and suggestions. This allowed us to introduce changes to the article which significantly improved the readability / clarity of our descriptions.
In the abstract, we added a record about the research hypothesis.
In the introduction - we added a research hypothesis and moved the fragment postulated for transfer to the discussion.
In the Materials and methods section, the map of the studied area with the location of the bypass
The Material and Methods section has been rebuilt (as suggested by another review)
Lines 188 - 210 were transferred to Conclusion and discussion
In the Discussion section, lines 298 - 305 were deleted.
Best wishes, Authors
Reviewer 2 Report
The manuscript “Changes in Land Plot Morphology Resulting from the Construction of a Bypass on the Example of the Polish City” presents an analysis of changes in land use (with emphasis on the area and shape of plots) before and after the construction of the Olsztyn bypass (in Poland).
My main concern about the manuscript is the lack of scientific soundness. The research design is affected by the vague information regarding the input data, the mentioned literature and the performed methodology.
Some major issues the authors should consider:
1. Cited literature: In the introduction section there are many relevant sentences regarding the manuscript topic, however most of them cited polish literature. Being the English the international language I think it is not acceptable that more than 25% of your references are in other language than English. I suggest the author to avoid so many articles in Polish and find other references that support, for example, line 38 (reference [2]), line 68 (reference [12]).
2. Morphology analysis applied: First, the authors mention there is only one shape indicator that relies on a land plot’s area and perimeter proposed by Kostrubiec (line 196-198), which as far as I know it is not true since a simple relationship between the area and the perimeter is as well a shape indicator. There is more indexes using area and perimeter. See for example the work of Limin Jiaoa and Yaolin Liu - ANALYZING THE SHAPE CHARACTERISTICS OF LAND USE CLASSES IN REMOTE SENSING IMAGERY. There is a lot of indexes that can give supportive information regarding the shape metric. Also, it would be more innovative to have a perception of the land plot defragmentation as the defragmentation have strong implications one the land plot meccanization and it production capacity, see the classic one of Fragstats.
3. Input data: The section 2.1. input data, it need significance content improvement. First, I would suggest to add a section regarding the study area and use part of information in the section 2.1. to describe the study area. Also, I cannot understand why the authors did not add a study area map. It is very hard to assimilate all the described information without one or more visual elements. In addition, there is no mention regarding the provenience of the used data to perform the analysis. What that data did you use? Where did you get the data? What are the used years? What are the data characteristics (e.g., Data model, Spatial representation, among others)? Also, the author should add a workflow representing the methodology to help the readers understanding the all the work that is proposed.
4. Distance values criteria: The authors should justify why they just used a distance of 400m and 800m (line 189) since according to the authors “the adverse consequences of motorway construction… are experienced within 2 km radius from the planned roadway” (line 66-68). Are these two distances values appropriate and sufficient for such analysis, there is no literature regarding this? A clear argumentation is missing.Also, I have some problems with the bold assumption made by the author (line 191).
5. I stopped reading the results section as soon as I realized that the formula for calculating the changes in the shape factor (line 273) is wrong. Did you calculated the changes based on the most recent mean area (ksr_an) minus the mean area in the past (ksr_a) divided by the most recent mean area? If so, is wrong, you should had divided by the area in the past (ksr_a). Also, the authors presented a formula without referring the meaning of each present element, what means ksr_an, ksr_a?
Due to lack of scientific soundness I suggest to revise all your results, conclusion and discussion.
Minor comments
Line(s) | Comment |
37 | Define what statistical data. |
38-39 | This information is insufficient to prove anything, 2016 may have been an atypical year. How was it in the other years? |
52-53 | Rephrase the sentences. |
63-64 | “…is usually analyzed” by whom? Author should clarify. |
112 | I suggest to define a priori what means land plot. I understood that for the authors land plot is just agricultural parcels? In other countries land plot can have other meanings. |
111-130 | I suggest to reorder this paragraphs, as you announced what you are presenting in the article, and only after is justified the importance of apply a morphology analysis. Also, there is no logical follow-up with the last paragraph. |
162-167 | I don’t see the relevance of this. |
168 | All this data is from what year? |
174-176 | The authors could add some figures to exemplifying. |
183 | What you mean with “…of the evaluated land plots were analyzed?” |
188-189 | As far as I can see, they are not parallel lines, the authors used a buffer! |
190-193 | The authors should rephrase the sentence. |
198 | I suggest to use other names for the plot’s area (Pdz) and perimeter (Odz) formula in order to be more easily to read. |
210 | The authors should redone the Figure 1. The distance lines are not legible, because they have the same color as the land plots areas. A scale bar and orientation are missing. Take off the name “legend:” it is not necessary. As it was made it looks like an island, the authors should add information around the road and the land plot (can be with transparence). Also, the distance lines are not right represent in the legend, it should be 0-400, 400-800, since like you have it means that the land plot intersected by the 400 meters line is also analyzed for the 800 meters line. |
Author Response
Thank you for your valuable comments and suggestions. This allowed us to introduce changes to the article which significantly improved the readability / clarity of our descriptions.
We would like to emphasize that in Poland in recent years, we observe increased work in the construction of new highways and expressways. Previous Polish research has focused on the effects of highway construction, mainly on agricultural land. The above fact is the reason for using of Polish literature. Agreeing with the suggestions of the reviewer, we added foreign literature.
Thank you for indicating a very valuable literature item, where shape indicators have been described. In our study, the selection of the shape index was based on the studies contained in Kostrubiec [52].
Data sources have been supplemented. A map with the research area has been added.
The revised article contains detailed explanations regarding the use of the 400 and 800 m lines (lines in the new version 202-210).
The formulas were corrected and detailed descriptions of the determinations were added. It should be emphasized that the studies assume that the plots crossed by the 800 m line from the road have not been disturbed by the location of the bypass and thus their shape has not changed. So, our base shape of plots are located 800 m away.
Other comments marked as Minor comments were included in a revised version of the article
Best wishes, Authors
Reviewer 3 Report
The manuscript describes the effects of a bypass road costruction on land parcel morphology, by means of 'Shape Factor'.
Even though the analysis might have some practical significance, there are different shortcomings that need to be addressed before the work could be considered for pubblication in "Sustainability" journal.
The main concerns are the following:
- The approach is based on the assumption that area and perimeter of land plots have a normal distribution over the region (lines 184-185). This assumption could be not correct since patch morphology throughout the region may depend on different factors. Hence, this assumption should be demonstrated by analysing a wider area of the region. The differences between land plots adjacent to the bypass vs. land plots of the surrounding should be then tested statistically to demostrate the effects of roadway costruction.
- Differences reported in tables 2 should be tested for statistical significance. The same when comparing values before vs after road costruction.
- The authors arbitrarly selected 400m and 800m as threesolds distances from the roadway. Why they used these distances? What they are based on?
- The Discussion section does not provide any discussion of the results. It sounds more like a literature review. Some discussion is instead provided in Conclusion section. Authors should also discuss implication on sustainability issue: for example, which are the environmental consequences of plot morphology changes?
Minor comments:
- Introduction section is very leghty and should be cutted significantly.
- Eq 2 and Eq 3: the different terms should be explicited and described in text
Author Response
Thank you for your valuable comments and suggestions. This allowed us to introduce changes to the article which significantly improved the readability / clarity of our descriptions. Referring to the reported major problems we would like to emphasize that the article does not deal with the environmental impact analysis because we did not have reflections on the implications of the morphology of the plots. We agree that this topic is interesting and requires research and should be analyzed in subsequent studies.
2. In our research, we do not make a cost comparison. In tables 2 and 3, we present statistical measures that allow you to compare the structure of plots before and after the location of the beltway.
3. In the revised version of the article, we have added an extended description justifying the adoption of a line away from the 400m and 800m road lanes. Generally, it should be emphasized that the plots crossed by the 800 m line from the road have not been disturbed (their shape has not changed) by the location of the beltway.
4. The discussion section has been rebuilt and combined with Conclusions. At the same time, we would like to emphasize that in our research we did not deal with the environmental effects of motorway construction, therefore it is difficult to look at the implications of the plots' morphology at this stage. There is no doubt that this topic should be considered in further studies.
Other minor comments have been fully taken into account.
Best wishes, Authors
Round 2
Reviewer 2 Report
The manuscript “Changes in Land Plot Morphology Resulting from the Construction of a Bypass on the Example of the Polish City” presents an analysis of changes in land use (with emphasis on the area and shape of plots) before and after the construction of the Olsztyn bypass (in Poland).
Overall comment
The paper was improved and the goal of the paper was better describe. Nevertheless, I still have major remarks concerning the methodology and some of the manuscript sections. The authors should see the MDPI “Instructions for Authors” and improve some sections. First, the abstract is not well structured. In addition, the introduction section it is too long and still does not provide to the readers in a clearly way, what are the objectives and what it is expected from this research. The section 2, still needs significance content improvement. The added maps are bad (no cartographic rules soundness), and there is still some gaps regarding the provenience the used input data. The equations used to analyze the changes are in the middle of results section. Lastly, I am not convinced about the methodology analysis applied (distance values used) and the land plots that are instersected by both buffers used. There is lack of scientific soundness and vague information regarding the equations used (e.g. equation 3 - the area is the intersection with the buffer or the all plot area?). The last section regarding the discussion and conclusions is still not appropriate. Additional experiments are needed to support the conclusions.
Reviewer 3 Report
Thanks for the revised version of the manuscript.
The text improved, but some more revisions are needed. My comments are the following:
Lines 239-254. You should be carefull presenting these differences. Looking at your data, it seems the differences have not statistically significance (S.D. is quite large). Stastical test (e.g. ANOVA) is strongly recommended.
you refer to "desiderable" shape factor (e.g. lines 173 and 333). What do you mean? desiderable values based on what?
lines 166-167 you confused "population" with "average plot" values.
line 199 "basing on literature analysis", please cite the sources
lines 352-360 here you describe possible impacts of changes in agricultral plots, including monetary values (farmers incomes related to parcel productivity). You could also mention impacts on other ecosystem services provided by agricultural land (with non-market economic values). For example, you can check and cite:
Power, A. G. (2010). Ecosystem services and agriculture: tradeoffs and synergies. Philosophical transactions of the royal society B: biological sciences, 365(1554), 2959-2971.
Gissi, E., Gaglio, M., Aschonitis, V. G., Fano, E. A., Reho, M. (2018). Soil-related ecosystem services trade-off analysis for sustainable biodiesel production. Biomass and Bioenergy, 114, 83-99.
You should also add few lines at the end of "conclusion and Discussion" section, with general conclusions of your study